# Surgery in Advanced Ovary Cancer: Primary versus Interval Cytoreduction

**DOI:** 10.3390/diagnostics12040988

**Published:** 2022-04-14

**Authors:** Mackenzie Cummings, Olivia Nicolais, Mark Shahin

**Affiliations:** 1Department of Obstetrics and Gynecology, Jefferson Abington Hospital, Abington, PA 19001, USA; mackenzie.cummings@jefferson.edu (M.C.); olivia.nicolais@jefferson.edu (O.N.); 2Asplundh Cancer Pavilion, Sidney Kimmel Cancer Center, Hanjani Institute for Gynecologic Oncology, Thomas Jefferson University, Willow Grove, PA 19090, USA

**Keywords:** advanced ovarian cancer, surgery, primary debulking surgery, neoadjuvant chemotherapy, interval debulking surgery, EORTC 55971, CHORUS, JCOG0602, SCORPION, overall survival

## Abstract

Primary debulking surgery (PDS) has remained the only treatment of ovarian cancer with survival advantage since its development in the 1970s. However, survival advantage is only observed in patients who are optimally resected. Neoadjuvant chemotherapy (NACT) has emerged as an alternative for patients in whom optimal resection is unlikely and/or patients with comorbidities at high risk for perioperative complications. The purpose of this review is to summarize the evidence to date for PDS and NACT in the treatment of stage III/IV ovarian carcinoma. We systematically searched the PubMed database for relevant articles. Prior to 2010, NACT was reserved for non-surgical candidates. After publication of EORTC 55971, the first randomized trial demonstrating non-inferiority of NACT followed by interval debulking surgery, NACT was considered in a wider breadth of patients. Since EORTC 55971, 3 randomized trials—CHORUS, JCOG0602, and SCORPION—have studied NACT versus PDS. While CHORUS supported EORTC 55971, JCOG0602 failed to demonstrate non-inferiority and SCORPION failed to demonstrate superiority of NACT. Despite conflicting data, a subset of patients would benefit from NACT while preserving survival including poor surgical candidates and inoperable disease. Further randomized trials are needed to assess the role of NACT.

## 1. Introduction

In the United States, it is estimated that in 2021, 21,410 women were diagnosed with ovarian cancer and 13,770 will die. Ovarian cancer is the fifth leading cause of cancer deaths and accounts for more deaths than any other cancer of the female reproductive system [1,2]. A woman’s risk of a diagnosis of ovarian cancer in her lifetime is about 1 in 78, and her risk of death due to ovarian cancer is about 1 in 108. There is increased frequency in diagnosis of ovarian cancer as age increases. About half of women diagnosed with ovarian cancer are over the age of 63 [2]. Up to 90% of malignant ovarian tumors are epithelial in origin [3]. Epithelial ovarian cancer (EOC) can then be further grouped into various histologic subgroups including serous cystadenocarcinoma (42%), mucinous cystadenocarcinoma (12%), endometrioid carcinoma (15%), undifferentiated carcinoma (17%), and clear cell carcinoma (6%). Overall prognostics factors of ovarian cancer include clinical stage, histologic grade, and residual amount of disease, and more recently homologous recombination proficiency [3].

Ovarian cancer (as well as fallopian tube carcinoma and primary peritoneal cancer) exhibits multiple modalities of metastasis. Direct invasion occurs when the cancer seeds into adjacent organs, most commonly the sigmoid colon. There is also subperitoneal spread which includes lymphatic, mesenteric, or hematogenous spread. Lymphatic spread includes extension of the cancer into the para-aortic, internal iliac, inguinal, and external inguinal nodes. Mesenteric spread occurs when the cancer infiltrates the broad ligament to involve structures such as the uterus, fallopian tubes, and contralateral ovary and may extend into the pelvic side wall and to non-contiguous organs. Hematogenous spread occurs when cancer enters into the ovarian or uterine vessels and subsequently travels to the liver, pleura, lung, adrenal glands, and spleen, with rare metastases to the bones and brain. Intraperitoneal spread occurs when cancer infiltrates the peritoneal cavity and spreads to areas such as the pouch of Douglas, right lower quadrant, terminal ileum, superior sigmoid mesocolon, right paracolic gutter, and the perihepatic recess. Peritoneal disease can also ascend through the diaphragm to the diaphragmatic, pericardial, and anterior mediastinal lymph nodes [4]. Given the many pathways for metastases, advanced ovarian cancer is often characterized by extensive disease with multi-organ involvement. 

The FIGO ovarian cancer staging (surgical) was most recently revised in January 2014 to better categorize disease states by observed prognosis or survival [5]. The most recent staging is outlined in Table 1. Significant changes included the elimination of stage IIC (IIA or IIB with positive washings) as well as expansion of stages IIIA and IV. Stage III is defined as cytologic or histologic spread to the peritoneum outside of the pelvis and/or metastasis to retroperitoneal lymph nodes. Stage IIIA was expanded from microscopic metastasis beyond the pelvis to IIIA1i (positive retroperitoneal lymph nodes only ≤1 cm), IIIA1ii (same as IIIA1i with metastasis >1 cm), IIIA2 (microscopic extrapelvic peritoneal involvement ± positive retroperitoneal lymph nodes). Stage IIIB was expanded beyond macroscopic extra pelvic peritoneal metastasis ≤2 cm to also include ± positive retroperitoneal lymph nodes or extension to capsule of liver/spleen. Stage IIIC was expanded beyond macroscopic extra pelvic peritoneal metastasis >2 cm to also include ± positive retroperitoneal lymph nodes or extension to capsule of liver/spleen. Stage IV is defined as distant metastasis excluding peritoneal metastasis. Stage IV was broken into two subtypes, IVA (pleural effusion with positive cytology) and IVB (hepatic and or splenic parenchymal metastasis, metastasis to extra abdominal organs including inguinal lymph nodes and lymph nodes outside of the abdominal cavity) [6]. Because the staging system was revised in 2014, comparisons of data prior to revision and post revision can sometimes be challenging.

The high mortality rate of ovarian cancer can be partly attributed to the high extent of disease diagnosed in advanced stages [7]. Across all types of ovarian cancer, 34% are stage III at diagnosis and 26% are stage IV at diagnosis. EOC is diagnosed at even more advanced stage with 37% stage III and 28% stage IV at diagnosis [8]. The late stage discovery of disease is often due to non-specific symptoms such as bloating, abdominal and pelvic pain, gastrointestinal symptoms including early satiety and change in bowel habits, and urinary symptoms, as well as a lack of beneficial routine screening.

Stage of ovarian cancer is known to be one of the most influential prognostic factors for ovarian cancer survival. The five-year overall survival for stage I EOC is as high as 90% and stage II is approximately 60%. In advanced stage (stage III and IV) disease, 5-year overall survival rates decline sharply to approximately 30%. As assessed using the Surveillance, Epidemiology, and End Results (SEER) registries, from 1983 to 2012, the overall incidence of ovarian cancer per 100,000 women decreased from 13.7 to 12.4 to 10.8 in the three decades included. Over the same years, 5-year survival increased over the 3 decades from 39.3% to 43.4% to 45.4%. Additionally, the mean survival improved from 34 months (mos.) to 46 mos. to 52 mos. [2,9].

The 2012 National Comprehensive Cancer Network (NCCN) guidelines for ovarian cancer stated the therapeutic benefit of neoadjuvant chemotherapy (NACT) followed by interval cytoreduction remained controversial, but may be considered in patients who were poor surgical candidates [10]. Before these recommendations, NACT was not recommended. This change was based on category 1 data from the European Organization for Research and Treatment of Cancer-Gynecological Cancer Group (EORTC-GCG) and the National Cancer Institute Canada-Clinical Trial Group (NCIC-CTG) data presented at the 2008 meeting of the International Gynecologic Cancer Society where overall survival was equivalent. However, the recommendation remained controversial as the reported overall survival of 29 and 30 months for each group was much less than the reported 50-month survival noted in randomized studies in the United States [10]. The recommendations for patients who should be considered for NACT are stage II–III disease, advanced age, and those with medical comorbidities who are not surgical candidates. In select patients, NACT followed by interval cytoreductive surgery could be considered where approximately 50% of patients will undergo complete resection. However, the recommendation remained in favor of primary cytoreduction and NACT could be considered only in cases with poor surgical candidates [10]. 

The current recommendations (2021 NCCN guidelines) for treatment of advanced FIGO stage III/IV ovarian cancer remain largely unchanged since 2012. The recommendation continues to be primary debulking surgery followed by platinum-based chemotherapy and a consideration for NACT in patients who are poor surgical candidates (i.e., advanced age, frailty, poor performance status, or comorbidities) or have a low likelihood of optimal cytoreduction [11]. While great progress has been made over the years with regard to surgical and medical management of ovarian cancer leading to a gradual increase in survival, primary treatment for advanced disease, namely primary debulking surgery (PDS) (cytoreduction) or NACT, remains controversial. In the United States, there has been an increased adoption of NACT following data from 2 randomized trials (Vergote, 2010; CHORUS, 2015) [12,13,14]. The rate of NACT use increased from 8.6% in 2004 to 22.6% in 2013 and from 17.6% to 45.1% from 2006 to 2016, but overall recommendations and clinical practice continue to remain in favor of primary cytoreductive surgery [12,15].

The purpose of this review is to summarize the evidence to date for primary cytoreductive surgery and neoadjuvant chemotherapy in the treatment of stage III/IV ovarian carcinoma in order to provide clarity in clinical decision making.

## 2. Materials and Methods

The database PubMed was systematically searched for relevant full text references using the search terms [(((ovarian OR ovary) OR fallopian tube OR primary peritoneal) AND (carcinoma OR cancer)) AND (primary debulking surgery OR primary cytoreduction)] and [(((ovarian OR ovary) OR fallopian tube OR primary peritoneal) AND (carcinoma OR cancer)) AND neoadjuvant chemotherapy] among humans, written in English. A total of 3566 articles were screened for relevant topics relating to advanced ovarian cancer (defined as stage III or IV) including (1) primary debulking surgery, (2) neoadjuvant chemotherapy, (3) diagnosis, (4) patient selection for NACT, (5) perioperative complications, and (6) adoption of NACT by physicians over time. Articles included randomized controlled trials, non-randomized prospective studies, retrospective studies, and review articles. Articles were screened by two authors (M.C. and O.N.) and included in the review if relevant to the previously mentioned topics. References of included articles were also screened and included if deemed relevant. Two authors (M.C. and O.N.) independently reviewed each article for relevant data. A schematic of the methodology can be visualized in Figure 1. 

## 3. Results

A total of 237 articles from the initial database search and screen of relevant references were included. Four randomized controlled trials comparing PDS to NACT followed by interval cytoreduction were found in the search and reviewed.

### 3.1. Primary Debulking Surgery

Primary debulking surgery (PDS) was first described in 1934 by Joe V. Meigs as a means to enhance the effects of radiation therapy prior to the development of modern chemotherapy [16]. Theories behind PDS as a treatment for ovarian cancer include the presence of a small population of highly specialized cancer cells responsible for tumor initiation, growth, and mutation that have the ability to self-renew and reengineer the entire cellular heterogeneity of a tumor. By removing these cells, regrowth of a therapy resistant tumor and recurrence is prevented. Another theory is that the number of cells treated with chemotherapy also decreases, thus decreasing the number of cancer cells that have the possibility to undergo spontaneous mutations to develop resistance during therapy [16,17].

With smaller tumor volume, there is also a higher possibility of tumor regression prior to the development of resistance. Removing large tumors may also increase sensitivity to chemotherapy as the remaining tumor cells would theoretically be more rapidly dividing and therefore targeted more effectively. Smaller residual tumors are better perfused resulting in higher growth rate of the tumor and more effective diffusion of chemotherapy agents into the tumor, increasing the efficacy of chemotherapy [17].

PDS was not widely accepted as the preferred treatment until the 1970s, when Griffiths published the first study demonstrating a direct relationship between residual tumor size and survival [18]. This was a retrospective, single institution study showing significant increase in overall survival if the residual tumor was less than 1.5 cm. The study concluded debulking surgery provides maximum benefit when all gross tumor can be resected, but there is limited utility in performing surgery if residual tumor is left behind [18]. This was quickly followed by smaller prospective and retrospective studies showing survival benefit of PDS, and the standard of care for advanced disease became PDS followed by chemotherapy [19,20,21].

There are several theories on what contributes most to survival in advanced ovarian cancers, emerging from investigations of PDS. These include size of residual disease following PDS, tumor biology, and surgeon/institution factors such as experience.

#### 3.1.1. Size of Residual Disease

Size of residual disease has been hypothesized to be the most important factor contributing to survival of advanced ovarian cancer. Over time, the size of residual disease considered optimal has been debated. In the 1970s, optimal residual disease was defined as ≤2 cm [22,23]. This was changed to <3 cm in the 1980s by Gynecologic Oncology Group (GOG) Protocol 47 followed by a change in 1986 to ≤1 cm in GOG 97, which was supported by smaller single institution retrospective studies showing increased survival at that threshold of residual disease [24,25,26]. Most recently, the goal of PDS has been no gross residual disease, but in cases where this is not possible, significant survival benefit is seen with residual disease ≤1 cm, which is considered optimally resected [24,27].

In 2002, Bristow et al. performed a meta-analysis of survival in stage III and IV ovarian cancer patients undergoing PDS followed by chemotherapy [22]. After controlling for other factors, the strongest predictor of median survival was found to be percent maximal cytoreduction where each 10% increase in cytoreduction was associated with an increase in survival of 5.5%, supporting the argument that the more disease removed at time of PDS, the longer the duration of survival [22].

In 2006, Chi et al. performed a retrospective single institution study of 465 patients from 1989 to 2003 diagnosed with stage IIIC epithelial ovarian carcinoma with the objective of analyzing survival at various residual disease diameters to determine the goal of optimal cytoreduction [24]. At the time of this study, the GOG threshold of residual disease was still <1 cm, and there was an effort to change this threshold to no gross residual disease. Results of this study revealed 3 groups with statistically significant differences in survival: no gross residual disease, ≤1 cm residual disease, and >1 cm residual [24]. In the no gross residual group, the median survival was 106 mos., which was one of the longest reported in patients with stage IIIC disease at the time [24]. In a previous study, in patients who were not optimally resected to microscopic or <1 cm of residual disease, there was survival benefit for residual disease <2 cm, but past 2 cm of residual disease, size does not affect prognosis [28]. However, in this study, beyond 1 cm of residual disease, there was no significant survival benefit from cytoreduction [24].

This study concluded removal of all gross residual disease had a significant improvement on survival and should be the aim of PDS. If this is not feasible, the goal should be cytoreduction to as minimal residual disease as possible as there may be an incremental increased survival benefit as extent of disease approaches no gross residual [24]. Although more extensive surgery resulted in increased survival, it also resulted in an increase in surgical complications [24].

Multiple other studies have also shown survival benefit with extensive surgery to achieve no gross residual disease [29,30,31,32]. The upper abdomen is frequently involved in advanced stage ovarian, tubal, and peritoneal cancers, and thus, extensive upper abdominal surgery as a means to achieve no gross residual disease would theoretically result in a survival benefit [33]. In one study, the authors analyzed survival of patients undergoing extensive upper abdominal surgery to achieve optimal residual disease of ≤1 cm compared to patients optimally debulked with standard surgical techniques and patients who were not optimally debulked [29]. All of the patients who were optimally surgically debulked had the same survival regardless of extent of surgery performed, implying that the presence of upper abdominal disease alone did not indicate poor tumor biology and initial maximum surgical effort improved survival in patients who otherwise would not have been optimally cytoreduced [29]. The group who were not optimally debulked had worse survival than the other 2 groups, thus supporting the effort of optimal cytoreduction to ≤1 cm residual disease [29].

Over time, a much more comprehensive approach has been taken to PDS resulting in significant improvements in rates of optimal cytoreduction and cytoreduction to no gross residual disease. Optimal cytoreductions were achieved in as little as 40–50% of cases prior to 2001 when extensive upper abdominal surgery became more routine [33,34]. Prior to 2001, patients with large volume upper abdominal disease involving the diaphragm, liver, or spleen were deemed unresectable and these patients were not optimally cytoreduced [34]. After extensive upper abdominal surgery became routine, the rate of patients with no gross residual disease as well as those optimally cytoreduced nearly doubled and concurrent survival benefits have been seen [33,34,35]. Given upper abdominal metastases are the most predictive factor for complete cytoreduction, and the need for at least one upper abdominal procedure has been seen in up to 50% of patients undergoing PDS (reflecting the proportion of patients with upper abdominal involvement), this paradigm shift was necessary to achieve complete cytoreduction in a larger proportion of patients [36,37]. 

Since the addition of extensive upper abdominal surgery, retrospective studies have shown continued increase in complete cytoreduction and also survival [38,39,40,41]. A single institution retrospective study showed an increase in complete gross resection, overall and progression free survival over a period of 13 years contributed to advancements in surgical technique to improve resection [42]. These practice-changing strategies have been shown in other studies to improve rates of complete resection as well as survival [43].

Due to survival benefit seen with no gross residual disease after PDS, the role of interval cytoreduction has also been studied, the theory being that patients who were not optimally cytoreduced at the time of initial surgery may be optimally reduced at a second surgery after receiving chemotherapy [44]. There has been no survival benefit seen in multiple studies analyzing secondary cytoreductive surgery [44,45,46].

Although PDS to no gross residual disease has been shown to have the greatest impact on survival, the risk of postoperative complications must also be considered. Factors such as age and functional status have been shown to increase risks of mortality and short term morbidity including sepsis, thromboembolic events, cardiac events, and reoperation [47,48]. Extensive surgery has been associated with increased operative time, increased estimated blood loss, and increased number of patients requiring blood transfusions postoperatively [34,35]. It has also been shown that women admitted emergently who underwent PDS were more likely to undergo bowel resection and had increased mortality compared to those admitted non-emergently [49].

Extent of surgery has been shown to be correlated with increased perioperative complications, but not mortality [34,47]. More complex surgery has been shown to have a survival benefit despite increased risk of complications, implying prognosis is more dependent on residual disease than age or other factors [47]. Additionally, the benefit of extensive surgery on OS has been shown to offset the risk of perioperative complications in multiple studies [50,51]. Despite the survival advantage associated with PDS, patient age, stage, functional status, and risk of complications must be considered when deciding which patients are appropriate candidates for PDS.

The goal of PDS to date remains no gross residual disease, but residual disease <1 cm is still considered optimal. No gross residual disease has the greatest prognostic impact on overall and progression free survival, but multiple retrospective studies have shown survival benefit of resection to <1 cm [52,53,54,55,56]. There may be worse survival associated with multiple sites of residual disease ≤1 cm as well as the presence of large volume ascites decreasing the chance of complete resection, but these have only been evaluated retrospectively [57,58]. Despite many factors concerning PDS including undertaking of extensive surgery and risk of complications, the consistent conclusion amongst the vast majority of studies is PDS should be undertaken whenever no gross residual disease is deemed at all possible due to impact on survival.

#### 3.1.2. Initial Disease Burden

Despite an abundance of data supporting increased survival with no gross residual disease following PDS, there have been other studies in the literature showing that initial disease burden and stage at presentation remains a prognostic factor even after achieving no gross residual disease [59]. It may not be the aggressiveness of the surgery that determines if complete cytoreduction is achieved, but may be the inherent tumor biology [60]. 

In 1992, Hoskins et al. performed a secondary analysis of GOG 52 to evaluate the influence of cytoreductive surgery on survival [23]. All of the patients enrolled in GOG 52 were cytoreduced to <1 cm [61]. The goal of this analysis was to determine if patients with large volume extra-pelvic disease who were optimally cytoreduced had the same survival as patients who were found to have extra-pelvic disease of <1 cm at baseline without extensive surgery [23]. Overall survival of patients with <1 cm extra-pelvic disease without debulking was 64 months (mos.) compared to 31 mos. in those patients who required debulking [23]. Because overall survival was significantly different between the groups despite the same diameter of extra-pelvic disease after initial debulking, it was hypothesized that factors in addition to extent of residual disease such as initial disease burden/stage may be more important for survival [23].

More evidence for this theory was provided by a secondary analysis of Scottish Randomized Trial in Ovarian Cancer (SCOTROC-1) performed in 2005 [62]. This study showed an inverse relationship between the volume of initial disease and progression-free survival despite optimal debulking to <1 cm residual disease in all patients enrolled in the trial [62]. Therefore, initial disease burden may be a major factor in survival and cannot be fully compensated for with more aggressive surgery [62]. 

Most recently, Horowitz et al. performed a retrospective study of patients enrolled in GOG 182 showing a survival benefit for no gross residual disease compared to those cytoreduced to <1 cm. However, this analysis also demonstrated shorter progression-free survival in patients with higher preoperative disease burden and surgical complexity was not an independent predictor of survival [59]. 

These studies contradict the data previously discussed showing extensive surgery to achieve no gross residual disease prolongs survival, and there may be a component of tumor biology and initial stage contributing to survival of advanced ovarian cancer patients that should be considered [24,29,30,31,34]. However, these studies were retrospective and compared overall survival across different stages, including stage IIIB and all stage III. These findings are also contradicted by multiple retrospective studies showing no relation of initial tumor burden, initial peritoneal dissemination, or site of tumor spread to survival [63,64,65]. Further prospective analysis comparing overall survival depending on disease volume within each stage is needed. 

#### 3.1.3. Surgeon/Institution Factors

Another factor that may be contributing to survival of advanced stage ovarian cancer patients is the surgeon and/or the institution performing the initial PDS. In the 2002 meta-analysis by Bristow et al., hospitals were divided into 2 groups: those specialized in cytoreduction where optimal resection was achieved in >75% of cases and those less experienced in cytoreduction where optimal resection was achieved in <25% of cases [22]. The results of this study demonstrated patients treated at a specialized center had an increase in survival by 50% [22]. This study also showed that only 20–40% of ovarian cancer patients had access to a specialized center [22].

A high degree of variability in aggressiveness of surgical effort to achieve no gross residual disease has been seen amongst gynecologic oncologists ranging from as low as 20% in less experienced centers to over 90% at more specialized centers [46]. In patients without access to a specialized center, there may not be opportunity for optimal debulking depending on surgeon availability. Additionally, women who are admitted emergently are more likely to be operated on in low volume hospitals by low volume surgeons and not at specialized centers [49].

Schrag et al. performed a SEER database study in 2006 to determine whether patients treated by high volume surgeons had better outcomes than those treated by low volume surgeons [66]. The results of this study showed the opposite of the previously mentioned studies, where there was no difference in overall survival between the two groups [66]. 

Bristow et al. performed a study in 2009 using the Maryland Health Service Cost Review Commission Database of 1894 patients who underwent PDS at 43 institutions performed by 352 surgeons. With a high volume surgeon, there was a 69% risk reduction of in hospital death and high volume hospitals were associated with increased likelihood of cytoreduction and shorter length of stay [67]. Multiple retrospective studies have also shown an increased likelihood of complete cytoreduction when PDS is performed at an experienced center, supporting these findings [68,69]. 

There is limited evidence assessing the role of specialization of the surgeon and the institution given all of the available data are retrospective, but patients operated on by specialists may have improved survival compared to those at low volume centers. The rate of optimal resection is higher at specialized centers and therefore this should be a consideration with PDS.

#### 3.1.4. Postoperative Chemotherapy

The standard of care for treatment of advanced stage ovarian cancer has remained PDS followed by adjuvant chemotherapy in patients with a high likelihood of achieving optimal cytoreduction who are good surgical candidates [70]. The effort to identify an effective chemotherapy regimen has resulted in improvement in survival over time with the emergence of platinum as well as intraperitoneal (IP) chemotherapy. In 2006, Armstrong et al. performed GOG 172, a randomized phase III clinical trial in patients with stage III epithelial ovarian cancer who underwent PDS with <1 cm residual disease [71]. Participants were randomized to 6 cycles of intravenous (IV) paclitaxel followed by IV cisplatin or IV paclitaxel followed by IP cisplatin and IP paclitaxel [71]. The median overall survival of the IV group was 49.7 mos. versus 65.6 mos. in the IP group. The IP group had more toxicity with only 42% completing all cycles of assigned IP therapy compared to 83% completing all assigned therapy in the IV group [71]. However, despite toxicity of the IP regimen, the IP group in this study had the longest median survival of all GOG randomized phase III clinical trials for advanced ovarian cancer to date [71].

The Japanese Gynecologic Oncology Group (JGOG) performed a multicenter randomized clinical trial (JGOG 3016) to investigate the effect of dose dense paclitaxel and carboplatin on PFS and OS in stage II–IV ovarian cancer [72]. The results of the initial analysis showed a significant improvement in PFS and OS in the dose dense regimen compared to the conventional regimen [72]. Long term analysis demonstrated significantly longer median PFS in the dose dense group of 28.2 mos. compared to 17.5 mos. in the conventional group and significantly longer OS of 100.5 mos. in the dose dense group versus 62.2 mos. in the conventional group [72]. Long-term adverse events were not assessed. The effect of treatment delays, dose reductions, and lower dose intensity of carboplatin were not prognostic of overall survival, but a lower relative dose density of paclitaxel was associated with a decrease in OS [72]. This study provided another possible chemotherapy regimen to prolong survival in patients with advanced disease. However, the outcome of this study was not able to be reproduced in the United States [73].

GOG 252 sought to identify which regimen is best between IP chemo and dose dense paclitaxel due to toxicities seen with both regimens and if bevacizumab should be added [73]. Following GOG 172, multiple studies were done to investigate IP chemotherapy regimens with decreased toxicity. GOG 9916 and 9917 substituted IP cisplatin with IP carboplatin and GOG 9921 decreased the dose of IP cisplatin [73]. GOG 218 demonstrated improved PFS with addition of bevacizumab [73]. Based on these studies, GOG 252 had the following arms: IV carboplatin, IP chemotherapy with carboplatin substitute, and IP chemotherapy with reduced dose of cisplatin, and all arms included IV paclitaxel and addition of bevacizumab including maintenance phase [73]. 

Median PFS in those with stage II–III that were optimally debulked to <1 cm residual disease was 26.9 mos. in the IV carboplatin arm, 28.7 mos. in IP carboplatin arm, and 27.8 mos. in the IP cisplatin arm. There was no statistical significance in survival between the arms for patients optimally debulked to <1 cm residual disease [73]. In patients with stage III ovarian cancer with no gross residual disease, median PFS was 35.9 mos. in the IV carboplatin arm, 38.8 mos. in IP carboplatin arm, and 35.5 mos. in the IP cisplatin arm. There was no statistically significant difference in progression-free survival in those with no gross residual disease after surgery [73]. All arms were associated with excessive toxicity, especially neurotoxicity, and efficacy may have been compromised by dose reductions as well as cross over between arms. However, the preliminary data from this analysis show dose dense paclitaxel may have improved efficacy and may be able to replace IP chemotherapy in the future [73].

### 3.2. Emergence of Neoadjuvant Chemotherapy

Despite the survival benefit seen with no gross residual disease or optimal debulking <1 cm after PDS, ovarian cancer remains one of the deadliest cancers in women and treatment strategies are far from ideal. NACT has been investigated as an alternative to PDS since the 1990s, especially in those patients with unresectable disease or those who are poor surgical candidates. However, the data remained largely retrospective until recent years. Despite multiple randomized controlled trials demonstrating survival benefit, NACT has only been included in NCCN guidelines for treatment of ovarian cancer since 2012, and significant controversy surrounding the data remains.

#### 3.2.1. Diagnosis of Advanced Ovarian Cancer 

Given tissue diagnosis of ovarian cancer most often occurs at time of surgery and staging is surgical, if NACT is to be considered, there must be a way to obtain a diagnosis of advanced ovarian cancer without performing PDS. The typical workup for a patient with signs and symptoms of ovarian cancer would be imaging, typically with transvaginal ultrasound or CT scan, and obtaining serum tumor markers such as CA125. While these tests can identify those patients who likely have ovarian cancer, NCCN guidelines recommend histologic confirmation of ovarian cancer (biopsy preferred) and/or laparoscopic evaluation to determine feasibility of resection prior to initiation of NACT [11]. 

Methods to confirm the diagnosis by cytology or histology include fine need aspiration (FNA), percutaneous biopsy, or diagnostic paracentesis. Cytology can be performed on FNA specimens as well as ascites obtained via paracentesis and has been shown to have a diagnostic accuracy of up to 98% [74,75]. A percutaneous biopsy of visual implants such as omental implants can be performed in order to confirm the diagnosis with histology, which has been shown to have diagnostic accuracy of up to 92% [74]. The benefit of these procedures is they can be done minimally invasively without the patient undergoing surgery. 

The role of laparoscopy in the diagnosis of ovarian cancer has also been investigated and is currently recommended by the NCCN as a means for obtaining a tissue biopsy of the cancer [11]. Imaging and laboratory findings can support a diagnosis of ovarian cancer, but there are other cancers and disease processes that can present with similar findings. In a retrospective analysis, it was cited that up to 7.1% of cases with a presumed diagnosis of ovarian cancer were actually incorrectly diagnosed with pathology consistent with metastatic uterine, breast, and gastrointestinal cancers [76]. Tissue confirmation can reduce inappropriate treatment as well as decrease inappropriate laparotomy. It has been shown that laparoscopy is relatively safe for this purpose with minimal blood loss and a short time from surgery to initiation of NACT [77].

Another consideration in the evaluation of suspected advanced ovarian cancer is to rule out advanced endometrial cancer, which is the most common gynecologic malignancy. This can be done with dilation and curettage or an endometrial biopsy. As previously mentioned, metastatic endometrial cancer can mimic advanced ovarian cancer and can be easily excluded with routine gynecologic procedures.

#### 3.2.2. Early Investigations of NACT

In the late 1990s, retrospective data began to emerge supporting NACT as an alternative in select patient populations. In 1998, Vergote et al. performed a retrospective study comparing NACT to PDS which showed a survival benefit in patients undergoing PDS who were optimally debulked, but survival was poor in patients with extensive stage IV disease despite PDS [78]. This study also showed a survival benefit in OS across all groups combined once PDS became more common in select patients, implying that there may be a role for NACT in certain patient populations with unresectable disease or comorbidities that may have led to postoperative complications if PDS was performed [78].

This was followed up by a retrospective study by Schwartz et al. in 1999 that compared NACT followed by interval debulking surgery (IDS) in a subgroup of patients versus PDS [79]. There was no difference in OS or PFS between the 2 groups, showing NACT was non-inferior to PDS. This study did not show the survival benefit that Vergote et al.’s study showed [78,79]. However, the patient population chosen to undergo NACT was statistically older with worse functional status than the group undergoing PDS, implying that NACT may be a reasonable alternative in patients who are poor surgical candidates and does not compromise survival [79]. The patient population chosen to undergo NACT remained consistent across early studies including those with multiple comorbidities or unresectable disease [80]. Following these studies, practice guidelines continued to recommend PDS for advanced stage disease as the preferred treatment, but NACT could now be considered if a patient was a poor surgical candidate [81].

During this era, the role for interval debulking following NACT was not well defined partly due to low usage of NACT. Multiple retrospective studies demonstrated a statistically significant benefit in OS in women able to undergo cytoreductive surgery following NACT than women who underwent NACT alone [79,82]. Additionally, the benefit of complete cytoreduction on overall survival was seen with interval cytoreduction, as previously demonstrated with PDS [83]. 

A non-randomized study investigated PDS versus NACT followed by IDS, where patients in the NACT group typically had worse prognosis or multiple comorbidities [84]. Optimal cytoreduction was achieved in a larger portion of patients in the NACT group and there was no difference in survival between the groups. Despite the non-randomized nature of this study, in patients with a worse prognosis or multiple medical comorbidities, NACT followed by IDS may not worsen prognosis, but may permit less aggressive surgery and improve quality of life postoperatively [84].

Another consideration is the optimal time to perform IDS after induction of chemotherapy. A meta-analysis in 2006 by Bristow et al. concluded that NACT in lieu of PDS resulted in inferior OS, but in patients selected for NACT, there was a negative effect on survival with increasing pre-operative chemotherapy cycles, suggesting that IDS should be undertaken as early as possible [85].

In the 2000s, further retrospective data emerged supporting NACT as a reasonable alternative to PDS. The selection of patients receiving NACT was commonly skewed towards those with high grade, more advanced stage disease, but optimal cytoreduction was achieved more often following NACT in this patient population [86]. Optimal cytoreduction remained the primary factor affecting median survival, so by increasing the rate of optimal cytoreduction following NACT, there would theoretically be an increase in median survival in this population that would have been otherwise sub-optimal if undergoing PDS [86]. Thus, it was concluded that NACT is a reasonable alternative to PDS in select patients with advanced disease.

Another emerging principle during this era was NACT as a means to decrease surgical morbidity without impacting survival. Multiple retrospective studies of patients with advanced stage disease demonstrated significantly less intraoperative blood loss and transfusions, shorter operating times, shorter hospital stays, and larger proportion of patients undergoing optimal cytoreduction in the NACT group with a similar survival to those undergoing PDS [87,88,89].

With emergence of retrospective data supporting NACT as an alternative to PDS in select patients, problems with NACT began to surface as well, especially lack of standardized way to determine patients appropriate for NACT versus PDS. In 2007, Bristow et al. performed a systematic review analyzing 26 studies for survival outcome achieved, degree of surgical effort/success, and selection criteria employed to select candidates for NACT [46]. This study concluded NACT was inferior to PDS but may be considered in patients felt to be optimally resected. Problems with NACT were highlighted including the need for standardized selection criteria that would be able to consistently and reliably detect patients in whom complete resection was unlikely without costing those with resectable disease the survival benefit of PDS [46]. The push for standardized selection criteria cited differences amongst surgeons’ willingness to employ extensive surgical efforts to achieve optimal cytoreduction. Given resectability was a subjective finding, patients may be deemed unresectable at some institutions and undergo NACT, but may have been selected to undergo PDS at another center [46]. With conflicting primarily retrospective data on effect on survival of NACT and no standardized way to choose appropriate patients to undergo NACT, the standard of care remained PDS with consideration for NACT in patients who were poor surgical candidates.

#### 3.2.3. Randomized Trials Investigating NACT

Prior to 2010, all data surrounding NACT were retrospective and no prospective randomized trials had been completed. Since then, four randomized controlled trials have been published comparing PDS to NACT, three of which were non-inferiority design and one superiority trial [13,14,90,91]. A summary of baseline characteristics of the patients included in each study can be visualized in Table 2.

In 2010, Vergote et al. published EORTC 55971, the first non-inferiority, randomized, prospective trial for patients with stage IIIC or stage IV epithelial ovarian, fallopian-tube, or primary peritoneal carcinoma comparing PDS versus platinum-based NACT followed by IDS [13]. The study ran from 1998 to 2006 and included 670 patients, 632 of which were eligible for randomization. Most patients had extensive disease with 74.5% having greater than 5 cm of metastatic disease and 61.6% of patients having greater than 10 cm of metastatic disease. All patients included in the study were required to have a World Health Organization (WHO) performance status of 0 (asymptomatic) to 2 (symptomatic and in bed for less than half of the day). The purpose of WHO performance status as inclusion criteria was that both groups of patients would have similar baseline characteristics, and all would be eligible for PDS or NACT. Patients were randomized to either PDS followed by at least 6 cycles of chemotherapy or 3 cycles of NACT followed by interval cytoreduction followed by at least an additional 3 cycles of chemotherapy [13].

Of note, prior to publication of GOG-152 in 2004, patients assigned to PDS who did not experience optimal cytoreduction were eligible to undergo secondary cytoreduction after completing adjuvant chemotherapy [13]. This was based on evidence from a randomized trial performed by Vergote in the 1990s that showed a survival benefit with PDS followed by chemotherapy followed by subsequent secondary cytoreduction over PDS followed by chemo alone [92]. However, after GOG-152 demonstrated no benefit in PFS or OS with secondary cytoreduction and thus failed to corroborate the previous study, this was no longer recommended [45].

The primary outcome of this trial was OS, and the secondary outcomes were adverse effects, quality of life, and PFS [13]. Overall survival was similar in the two groups with median overall survival of 29 mos. in the PDS group and 30 mos. in the NACT group. The median progression-free survival in both groups was 1 year. OS for patients who underwent IDS after suboptimal primary debulking was similar to the NACT group. Subgroup analysis with regard to age, FIGO stage, WHO performance status, histologic type, and presence or absence of pleural fluid was performed to identify patients who would benefit from one treatment over another, but did not demonstrate any difference between treatment groups. The strongest independent variable of OS was complete resection of disease, regardless of if this was completed at primary or interval surgery. Of note, 80.6% of the patients in the NACT group were optimally resected to ≤1 cm at IDS versus 41.6% of patients in the PDS group. Primary adverse effects and mortality were also higher in the PDS group [13]. 

One major critique of this study is overall poorer outcomes with regard to OS and PFS compared to previously reported survival data. This trial included patients that had bulky stage IIIC or IV disease with 61.7% of patients having >10 cm of metastatic disease and did not include those with stage IIIB or earlier [13]. The overall survival in this study was compared to other single institution series of which two included only stage IIIC, two included all stage III and IV, and one included stage IIIC and IV only but in the setting of advanced diaphragmatic resection with the goal of achieving optimal primary cytoreduction [22,24,31,33,78]. Given the inclusion of only those with stage IIIC and IV disease, the population of this trial was theorized to have a worse prognosis at baseline compared to other single institution series. OS and PFS reported in this trial were similar to other regional and multicenter studies that focused on survival in stage IIIC and IV specifically [32,62,66,93,94,95,96].

Overall, this trial confirmed complete resection of all macroscopic disease is the most important prognostic factor, but this end goal may be achieved by either by NACT with interval debulking or primary cytoreductive surgery [13]. The decision to proceed with NACT versus primary debulking surgery is a multifactorial decision and should include an assessment of factors such as coexisting illnesses, age, disease burden WHO performance status, tumor stage, and location of metastatic sites. While previous practice would tend to favor NACT only for patients who were not surgical candidates due to some of the aforementioned factors, this study opens the door to a potential greater consideration of NACT in other populations, as non-inferiority was demonstrated within a population, all of whom were surgical candidates. Additionally, NACT followed by IDS was advantageous in achieving optimal reduction more often and was associated with a lower postoperative mortality, shorter operation times, less grade 3 hemorrhage, fewer venous complications, and fewer infections [13].

In 2015, Kehoe et al. published data from CHORUS, the second randomized, controlled, non-inferiority study investigating primary chemotherapy versus primary surgery for advanced ovarian cancer [14]. The trial was designed in accordance with the previously discussed EORTC 55971 trial with the intention of combining the results in a meta-analysis. This study highlighted the need for an alternative to PDS, the standard of care at the time, as more than 75% of women diagnosed with ovarian cancer were stage IIIC or IV with a large proportion unfit to undergo surgery and have a less than 25% 5-year survival rate. The purpose of this study was to test the hypothesis that NACT could result in survival similarly to primary debulking surgery with a reduction in surgical morbidity [14].

CHORUS was a multicenter, randomized phase 3 trial completed in the UK and New Zealand of all stage III (compared to stage IIIC only in EORTC 55971) or IV ovarian, fallopian tube or primary peritoneal cancer [13,14]. All patients included in this study were candidates for either randomization arm. Patients were randomized to either platinum-based NACT or PDS. Similarly to EORTC 55971, the PDS group would undergo surgery followed by 6 cycles of platinum-based chemotherapy and the NACT group would receive 3 cycles of chemotherapy followed by IDS followed by an additional 3 cycles of completion chemotherapy. The primary outcome of this study was overall survival. Secondary efficacy outcomes included progression-free survival and quality of life [14].

The study was conducted from 2004 to 2010 and included 550 women in the final analysis. A total of 276 women were assigned to primary surgery and 274 were assigned to NACT with similar baseline characteristics between groups. There was no difference in survival between both groups, but survival was lower than expected across the entire cohort. Mean overall survival was 22.6 mos. in the PDS group and 24.1 mos. in the NACT group. The findings demonstrated non-inferiority of survival with NACT compared to PDS. Again, optimal cytoreduction to <1 cm was achieved in a higher proportion of the NACT group (73%) compared to the PDS group (41%). Subgroups such as age, stage, tumor size, performance status, and planned chemotherapy did not show any subgroup benefited more from either treatment arm [14].

The NACT group in this trial had better scores on quality of life assessments at 6 mos. and 12 mos. time points post treatment. The NACT group also had fewer adverse events, shorter hospital stays, and fewer postoperative deaths within the first 28 days than the primary surgery group [14].

While this study did differ in regard to stage inclusion, a large percentage (88%) were FIGO stage IIIC or IV making the population quite similar to EORTC 55971. Similarly to EORTC 55971, CHORUS was criticized due to lower expected median OS across both arms of the trial [14]. It was hypothesized that lower survival than expected could be due to older median age, 77% of tumors were poorly differentiated, and a higher percentage of women (19%) had poorer performance status compared to previous studies. This study concluded that NACT is a reasonable alternative to PDS and is non-inferior with regard to survival [14].

EORTC 55971 and CHORUS were then combined and analyzed together in a meta-analysis. The aim of this analysis was to show non-inferiority in OS with NACT compared with PDS [97]. Data for 1220 women were included, the majority of which had stage IIIC (86%) or IV (19%) disease. Median follow up was 7.6 years for patients included in EORTC 55971 and 5.9 years for patients included in CHORUS. When combined, there was no difference in median OS between NACT and PDS, 27.6 mos. and 26.9 mos., respectively. A subgroup analysis of women with stage IV disease was completed and statistically significant increase in OS was seen with NACT (24.3 mos.) compared to PDS (21.2 mos.). This analysis concluded that NACT is non-inferior to PDS in stage IIIC–IV ovarian, fallopian tube, or primary peritoneal cancer with long term follow up and there may be a survival benefit of NACT over PDS in patients with stage IV disease. NACT was concluded to be a reasonable alternative for patients with advanced stage ovarian cancer, especially those with high tumor burden or poor preoperative performance status [97].

The Japan Clinical Oncology Group (JCOG) performed a phase III randomized clinical trial, JCOG0602, to first assess invasiveness of surgery with NACT followed by IDS versus PDS followed by an analysis of OS and PFS [90,98]. The study was a randomized, open-label phase III non-inferiority trial in 34 Japanese centers conducted from 2006 to 2011. A total of 301 patients with stage III or IV ovarian, tubal, or peritoneal cancers were randomized to PDS followed by eight cycles of platinum-based NACT or 4 cycles of platinum-based NACT followed by IDS followed by four more cycles of NACT. In the PDS arm, IDS was optional for patients with suboptimal PDS [90,98].

The first analysis of this trial was to compare treatment invasiveness between the groups. This analysis found the NACT arm had less surgical requirement, shorter operation time, lower frequency of abdominal organ resection, lower distant metastases resection, smaller blood/ascites loss, decreased frequency in albumin transfusion, and decrease in grade 3 or 4 adverse events. Thus, the study concluded IDS following NACT was less invasive than PDS [98].

The study’s final analysis included OS and the major secondary outcome PFS [90]. Baseline characteristics were similar between groups. Optimal cytoreduction was achieved in 12% of patients in the PDS group and 64% in the NACT followed by IDS group. Of note, 49 of 147 patients who originally underwent PDS also underwent subsequent IDS and optimal resection was achieved in 31% of these patients, resulting in a total of 37% of patients in the PDS group undergoing optimal resection. Of note, this trial started after GOG-152 was published which showed no benefit in secondary debulking after PDS on survival [45]. Median OS was 49 mos. in the PDS arm and 44.3 mos. in NACT arm, but this was not statistically significant (*p* = 0.24). Median PFS was 15.1 mos. in the PDS arm and 16.4 mos. in the NACT arm, but this was also not statistically significant [90]. 

This trial failed to show non-inferiority of NACT followed by IDS compared to PDS and thus disagreed with the results of EORTC 55971 and CHORUS. Criticisms of this trial include a smaller sample size resulting in lower statistical power. Another consideration was a large proportion of the PDS group (33%) underwent IDS and this was done less often (17%) in EORTC 55971 and not described at all in CHORUS due to the data from GOG-152 [90]. However, as previously mentioned, a study performed by EORTC did show a survival benefit from IDS following PDS and a Cochran systematic review and meta-analysis showed benefit of interval debulking in patients who underwent PDS performed by non-gynecologic oncologists, i.e., those who underwent suboptimal resection [92,99]. Given the high rates of suboptimal cytoreduction in PDS, it was determined there may be benefit from PDS followed by IDS in this study [90].

This was followed by publication of the SCORPION trial by Fagotti et al. in 2020. This trial considered the previous randomized studies investigating NACT versus PDS, EORTC 55971 and CHORUS, and instead, chose to focus on different primary outcomes, perioperative complications, and progression-free survival [91]. While non-inferiority of NACT had been shown in EORTC 55971 and CHORUS, these trials had limitations including a wide range of disease stage and performance status in the included patients, as well as low optimal cytoreduction rates in the PDS groups. Primary treatment modality of NACT or PDS may have different efficacy in patients depending on the extent of disease and other baseline characteristics, and the heterogeneous nature of the populations in these two trials may have skewed the results [13,14,100]. 

SCORPION aimed to overcome these limitations in analysis and interpretation. This was a randomized, single institution open-label phase III superiority trial. Eligible patients were aged 18 to 75 with stage IIIC or IV ovarian, fallopian, or primary peritoneal cancer, an Eastern Cooperative Oncology Group (ECOG) performance status of 0 to 2, and no previous history of chemotherapy [91]. Additionally, similar to the previously discussed trials, patients had to be candidates to undergo either NACT or PDS in order to allow for appropriate randomization. Patients underwent laparoscopy to assess extent of disease and those with high tumor burden were randomized to NACT or PDS. Tumor burden was determined by the Fagotti score, which is a standardized laparoscopic predictive index designed to predict probability of optimal cytoreduction in advanced ovarian cancer patients. The score takes into account peritoneal carcinomatosis, liver metastasis, and involvement of the diaphragm, mesentery, omentum, bowel, or stomach. Higher scores indicate decreased probability of optimal cytoreduction [101]. Of note, this was the only trial to use laparoscopy as a means to assess disease burden prior to randomization [91].

The primary outcomes were superiority of NACT versus primary debulking surgery in terms of perioperative morbidity and PFS [91]. These primary outcomes were chosen based on findings from EORTC 55971 which suggested NACT may be superior in regard to clinic outcome in patients with stage IV disease and PDS may be superior in regard to survival for patients who initially presented with disease burden of <5 cm [13]. Secondary outcomes were OS and quality of life assessment [91].

A total of 171 patients were included in the study, 84 of which were assigned to PDS and 87 were assigned to NACT. Randomization was not stratified by patient characteristics, but there were no significant differences in baseline characteristics between the two groups. After completion of the study, 71 patients underwent PDS and 72 patients completed NACT [91].

There was a significant difference between rates of complete resection between the PDS group (47.6%) and the NACT group undergoing IDS (67%). Additionally, the extent of surgery as expected was higher in the PDS group. Similar to findings in the EORTC 55971 trial, operating times and hospital stays were significantly longer for the PDS group. Additionally, more of the major postoperative complications occurred in the PDS group with death due to postoperative complications in 8.3% of the PDS arm [91]. There was no statistically significant difference in PFS between the groups, 15 mos. in the PDS group compared to 14 mos. in the NACT group. There was also no statistically significant difference in OS between the groups, 41 mos. in the PDS group and 43 mos. in the NACT group. These reported median overall survival rates were more in line with the expected survival when compared to the overall survival rates reported in the EORTC 55971 and CHORUS trials. This difference in survival may be due to a younger and higher performance status population when compared to the EORTC/CHORUS trials. On multivariate analysis of patient characteristics such as age, performance status, CA125, and residual tumor at surgery, only residual tumor at surgery and CA125 had independent prognostic value [91].

The results of this study demonstrated NACT followed by IDS resulted in significantly lower postoperative complication rates than PDS. However, they failed to show superiority of NACT for OS or progression-free survival compared to PDS. Overall, complete cytoreduction remained the top prognostic factor. It may be extrapolated that the NACT group had the added benefit of a higher proportion of patients with complete cytoreduction after surgery when compared to the PDS group. However, there are many factors such as primary chemo-resistant patients that may confound this theoretical benefit as seen by the lack of significant difference between OS and progression free survival between groups. This study did have several limitations including lack of statistical power to detect a minimal difference in months of survival, the single center design which can be difficult to generalize to the larger population, and a much smaller cohort than the previous trials [91].

The conflicting data presented in these four trials make it difficult to translate into meaningful clinical practice. While EORTC 55971 and CHORUS showed non-inferiority of NACT versus PDS with respect to OS, JCOG0602 failed to show non-inferiority [13,14,90]. SCORPION was then designed completely differently and was a superiority trial with PFS as a primary outcome instead of the previous design of non-inferiority with OS as the primary outcome. SCORPION failed to demonstrate superiority of NACT with respect to PFS or OS [91]. Given the differences in trial design, it is difficult to compare the results of these studies. A comparison of trial design can be visualized in Table 3. 

Interestingly, despite conflicting data with regard to survival, comparison of postoperative complications and deaths between PDS and NACT followed by IDS were consistent amongst the four trials. All four trials demonstrated more postoperative complications and deaths in the PDS group compared to the NACT group [13,14,90,91]. The rate of postoperative deaths in EORTC 55971 was 2.5% in the PDS group compared to 0.7% in the NACT group [13]. The rate of postoperative deaths in CHORUS was 6% in the PDS group compared to <1% in the NACT group [14]. The rate of postoperative deaths in JCOG 0602 was very low across all participants compared to the previous two trials with 0.7% in the PDS group and no postoperative deaths in the NACT group [90]. The rate of postoperative deaths in SCORPION was 8.3% in the PDS arm and no postoperative deaths in the NACT group [91]. The four trials may not provide a consensus on survival benefit of NACT, but consistently demonstrated decreased frequency of postoperative adverse events of those undergoing NACT and IDS.

#### 3.2.4. Conflicting Data for NACT

The conclusions drawn by these trials have also been criticized and therefore have not been widely accepted by clinicians. During this era, a retrospective cohort study using the National Cancer Database (NCDB) was also published comparing OS in PDS versus NACT due to criticism and lack of acceptance of the non-inferiority of NACT shown by EORTC 55971 and CHORUS [102]. The study included stage IIIC or IV epithelial ovarian cancer between 2003 and 2011. All patients included were <70 years old with Charleson Comorbidity Index (CCI) of 0 and were likely candidates for either treatment. PDS was associated with longer median OS of 37.3 mos. compared to 32.1 mos. in NACT group. Of note, of the 22,962 patients included, only 3126 received NACT. Factors associated with use of NACT included older age at diagnosis, more recent diagnosis, serous histology, and stage IV disease. After propensity score matching, the baseline characteristics were well balanced between the 2 groups, effectively eliminating these factors as contributory to the difference in OS. However, one difference between the groups was the NACT group had a higher proportion of women with ECOG performance scores of 1–2 than the PDS group, which may have contributed to the difference [102].

Additional retrospective studies using national databases have shown conflicting results. Seagle et al. performed a retrospective study using the NCDB showing better OS in the PDS group in women with stage III disease, but this difference was not seen in those with stage IV disease [103]. Additionally, Lyons et al. performed a retrospective study using the NCDB showing better OS in the PDS group after adjusting for age, comorbidities, stage, and residual disease, and actually showed an increase in 30 day mortality in the NACT group [104]. Mysona et al. also performed a study using the NCDB database showing better survival with PDS than NACT, but this only remained true if PDS resulted in optimal cytoreduction [105]. A study performed using the SEER database showed better survival in the PDS group, but fewer surgical complications in the NACT group [106]. Another study performed using the SEER database also showed better survival with PDS, but only in stage III patients and no difference in survival in stage IV patients or those over age 80 [107].

In addition to large retrospective studies using national databases, our search yielded conflicting data from retrospective single institution studies comparing PDS to NACT. Of these studies, 13 showed no difference in survival between NACT and PDS [108,109,110,111,112,113,114,115,116,117,118,119,120,121,122,123], 16 studies showed better survival in the PDS group [124,125,126,127,128,129,130,131,132,133,134,135,136,137,138,139], and only 2 studies demonstrated superiority of NACT to PDS [140,141]. Our search yielded one prospective observational study showing better survival in the PDS group than NACT and one multi-institutional retrospective study showing no difference in median OS but a higher risk of death at 2 years in the NACT group [142,143]. Of the meta-analyses yielded in our search, five showed no difference in survival of PDS versus NACT [144,145,146,147,148,149] and two showed PDS with improved survival relative to NACT [146,150].

NACT has been shown in the majority of studies to result in complete cytoreduction more often than PDS, but concurrent survival benefit has not been shown, with most studies citing no difference in survival or worse survival with NACT [88,105,111,112,114,115,121,128,138,148,149,151]. Rare studies have shown that NACT does not improve the rates of optimal cytoreduction, but these represent a minority of studies [152,153]. 

In the setting of conflicting data for survival with NACT compared to PDS, other considerations including perioperative outcomes need to be addressed. In the vast majority of studies, NACT followed by IDS has been associated with fewer surgical complications, decreased surgical mortality, lower estimated blood loss, lower volume of ascites, shorter operation time, shorter hospitalizations following surgery, and improved quality of life postoperatively [38,105,107,117,141,154,155,156,157]. PDS has been associated with higher surgical complexity, a higher rate of grade 3 or 4 surgical complications, postoperative infections, wound complications, vascular events, ICU admissions, reoperations, and readmissions [110,120,125,146,155,158,159]. The greatest benefit from PDS that outweighs risks of these complications are in patients with lower surgical complexity and without pre-existing comorbidities [160]. Rare studies have shown no difference in perioperative complications or even higher postoperative mortality, but these are by far the minority [104,161]. One single institution retrospective study found patients undergoing NACT followed by IDS had a higher rate of perioperative blood transfusions, but this was thought to be due to worse baseline anemia following NACT [162]. PDS has also been associated with an increase in bowel surgery including increased rates of rectosigmoid resection, but this is not necessarily correlated with an increase in ostomy formation [107,163,164].

Despite early retrospective data showing benefit of NACT over PDS and randomized trials demonstrating non-inferiority of NACT, NACT has not been widely accepted due to conflicting data. Conclusions that have remained relatively consistent across studies are that NACT followed by IDS leads to less postoperative morbidity and is associated with higher rates of complete resection. Another consistent point is the importance of baseline characteristics of the patient population chosen to undergo NACT versus PDS. Most studies at this point recommend PDS for all patients with possible optimal cytoreduction and to reserve NACT for those with inoperable disease or comorbidities precluding them from surgery.

#### 3.2.5. Timing of Interval Cytoreduction 

An additional consideration is there is limited prospective data regarding the timing of IDS after NACT. All four randomized trials tested surgery following 3–4 cycles of NACT in patients who had a response to NACT and were stable [165]. Data from retrospective studies have been conflicting. Several studies have shown no difference in survival if IDS is delayed past three cycles of NACT, especially if complete resection is more likely to be achieved with more than three preoperative cycles [166,167,168,169,170,171,172]. However, there have also been studies showing worse survival with NACT followed by IDS with >4 cycles of preoperative chemotherapy despite optimal cytoreduction at time of IDS [173,174]. Additionally, studies have shown that platinum resistance increases after three cycles and patients who have had a complete response after three cycles do not benefit from further preoperative chemotherapy [175,176]. CA-125 levels may also be useful in guiding timing of IDS as normalization of CA-125 as well as absence of ascites prior to IDS may improve rates of complete cytoreduction and survival [177,178]. It has also been shown in one study that the number of preoperative chemotherapy cycles prior to IDS does not have any effect on improving survival compared to PDS [169].

The current recommendations are for patients receiving NACT to undergo surgery after four or fewer cycles of chemotherapy. Alternative timing of surgery has not been prospectively studied, but may be considered on an individual patient basis based on CA-125 drawn each cycle and early performance of radiographic imaging to assess disease response [165].

#### 3.2.6. Problems with NACT

While several prospective studies have demonstrated non-inferiority of NACT to PDS for advanced ovarian cancer in certain patient populations, several retrospective studies have cited problematic outcomes associated with NACT. Several studies have focused on outcomes among patients treated with PDS versus NACT such as disease recurrence and platinum resistance (defined as recurrence of disease < 6 mos. after treatment with platinum-based therapy). The underlying theory to these investigations is that by exposing large tumor volumes to chemotherapy (i.e., with NACT), the risk of proliferation of drug-resistant tumor cells and platinum chemotherapy-resistant disease increases.

After EORTC 55971 showed non-inferiority of NACT compared to PDS with regard to survival, Rauh-Hain et al. performed a study analyzing the effects of NACT on the development of platinum-resistant disease as these patients have earlier chemotherapy exposure compared to patients undergoing primary debulking surgery [179]. This was a retrospective cohort study including 425 patients with stage IIIC and IV EOC, where 22.3% of the patients underwent NACT-IDS and 77.6% underwent PDS. On univariate analysis, there was a statistically significant increase in the number of patients who were found to have platinum-resistant disease after initial platinum-based chemotherapy in the NACT with IDS group compared to the PDS group (44.2% versus 31.2%, respectively) [179]. However, on multivariate analysis, this difference was no longer observed. Stage of disease, sub-optimal cytoreduction, and more than six initial cycles of platinum based chemotherapy remained the only factors affecting platinum resistance [179].

In patients who had a first recurrence that was treated with platinum based chemotherapy, a higher percentage of women who had been initially treated with NACT and IDS compared to PDS went on to develop a second, now platinum-resistant, recurrence within six months (88.8% versus 55.3%, respectively) [179]. This study also suggests increased cycles of platinum chemotherapy increases the risk of platinum-resistant disease and therefore, IDS should be based on response to chemotherapy rather than a set number of cycles in order to reduce the risk of platinum resistance in recurrent disease [179].

In 2013, Petrillo et al. published another study analyzing the timing and pattern of recurrence in advanced stage ovarian cancer patients treated with PDS versus NACT followed by IDS, as well as platinum resistance in these groups [100]. A total of 175 patients with stage IIIC–IV epithelial ovarian cancer were included in this study where 22.9% of patients underwent PDS and the remaining 77.1% underwent NACT with IDS. Compared to the study by Rauh-Hain et al., this population had a much higher percentage of patients who underwent NACT with IDS compared to PDS [100,179]. All patients included had no residual tumor at the end of treatment, and there were no significant or pathologic differences between the two groups. There was a statistically significant difference in recurrence rates where 50% recurred in the PDS group and 76.3% recurred in the NACT with IDS group [100]. This study also showed a statistically significant increase in platinum-resistant recurrences as well as a shorter interval to platinum resistance in the NACT group compared to the PDS group. This study concluded that while rates of optimal debulking may be higher in certain studies in the NACT groups, the implications on recurrence must be considered when choosing a therapy plan [100].

After publication of both EORTC 55971 and CHORUS, da Costa et al. demonstrated similar findings regarding platinum resistance after NACT with IDS [180]. This retrospective cohort study included 237 patients with stage IIIC and IV ovarian carcinoma where 62.0% of patients were treated with PDS versus 38.0% NACT with IDS. Of note, patients in the PDS group were younger, had fewer rounds of chemotherapy, and were more likely to have residual disease <1 cm. The time to platinum-resistant relapse was much longer in the PDS group (80.8 months) compared to the IDS group (39.9 months) as well as those with residual disease <1 cm compared to those with residual disease >1 cm [180]. This study suggests the overall response rate for patients treated with platinum-based chemotherapy in recurrent disease was worse for patients treated initially with NACT followed by IDS compared to PDS, but this was not statistically significant. Overall, this study concluded NACT followed by IDS was associated with a shorter time to development of platinum-resistant disease [180]. Multiple other retrospective studies have also shown increasing platinum resistance with NACT [123,181].

While there are differences in the results of these retrospective analyses, the findings suggest NACT may contribute to platinum resistance, higher risk of a platinum-resistant recurrence, and shorter interval to development of platinum resistance. However, these claims have never been assessed in a prospective study. The decision on which patients should undergo NACT versus PDS remains challenging, but the possibility of platinum resistance with NACT should be considered.

#### 3.2.7. Candidates for NACT

Prior to the emergence of any data supporting NACT as a reasonable alternative, a small cohort of patients underwent NACT as primary treatment for a variety of reasons, leading to investigations into patient populations who would benefit most from NACT. Patient and institution characteristics that have historically been associated with choosing NACT over PDS include older age, frailty, serous or unclassified histology, stage IV disease, greater disease extent, insurance status, distance from treatment center, treatment center adoption of NACT, and treatment center adherence to NCCN guidelines [168,182,183,184,185,186,187].

If conclusions drawn from EORTC 55971 and CHORUS are to be accepted and NACT is non-inferior to PDS, there must be a way to select which patients are appropriate for which treatment. A secondary analysis of baseline characteristics from patients enrolled EORTC 55971 was performed in order to identify a subgroup of patients that would benefit most from PDS or NACT [188]. This analysis identified size of the largest metastatic tumor and clinical stage had a significant association with benefit from treatment. Patients with stage IIIC disease with metastatic tumors ≤45 mm benefited more from PDS in terms of 5-year survival, while patients with stage IIIC disease with metastatic tumors >45 mm and patients with stage IV disease benefited more from NACT followed by IDS in terms of 5-year survival. Size of metastatic disease and clinical stage therefore may have an impact on benefit from treatment modality and should be considered when choosing primary therapy [188].

The survival benefit of NACT in patients with bulky metastases or stage IV disease may be due to increased likelihood of complete resection at time of IDS following NACT. As previously discussed, complete resection of all gross residual disease continues to be the most influential factor on survival and it has been shown in multiple trials as well as retrospective analyses that optimal resection is achieved in a higher proportion patients undergoing NACT followed by IDS than in those undergoing PDS [13,14,90,189].

It can be difficult to assess the likelihood of optimal debulking and several techniques including laparoscopy, imaging, CA-125, and patient factors have been proposed as a means to predict optimal cytoreduction. The problem with these predictive algorithms is the heterogeneity of the studies with most not externally validated.

LapOvCa, a randomized control trial of 201 patients out of the Netherlands, demonstrated a reduction in the number of futile laparotomies in patients with suspected advanced stage disease with use of diagnostic laparoscopy, but this was based on a series of largely subjective criteria assessed by the surgeon [190]. In other retrospective studies, laparoscopy has been shown to be fairly accurate in predicting complete cytoreduction and was not associated with adverse surgical outcomes [191,192,193,194,195]. Laparoscopy may also be more accurate than CT scan to assess carcinomatosis in pelvic and small intestinal regions [196]. Despite studies showing accuracy of diagnostic laparoscopy to predict complete cytoreduction no impact on survival has been seen, but can be useful to reduce unnecessary laparotomies [197].

Several standardized scoring systems such as the Fagotti score, R3 and R4 models, and Sugarbaker’s peritoneal cancer index (PCI) have been developed as predictive models using laparoscopy to assess likelihood of optimal cytoreduction [101,198,199,200]. The Fagotti score assesses several features at the time of laparoscopy as potential predictors of surgical outcome including presence of ovarian masses (unilateral or bilateral); omental cake with tumor spread to the lesser and greater curvature of the stomach; diaphragmatic, peritoneal, bowel, and liver carcinomatosis; large and/or small bowel infiltration; and mesenteric retraction [101,199]. The R3 model score is obtained from a combination of preoperative computed tomography of the chest and abdomen, calculation of laparoscopic PCI, and presence of clinically or radiographically diagnosed partial bowel obstruction. The R4 model adds operative PCI to the R3 model [199]. 

Sugarbaker’s PCI is a summation of peritoneal implant size scored based on the largest implant and distribution of implants in 13 designated abdominopelvic regions to assess likelihood of complete cytoreduction in patients with extensive peritoneal surface malignancy [200]. Conflicting data regarding a score for PCI predictive of suboptimal resection have been shown in non-randomized prospective and retrospective studies with most studies quoting PCI > 20–24 as the threshold when NACT should be considered [196,201,202]. While these tools have been validated in some studies and can provide helpful information, they are not comprehensive with regard to the entire clinical picture and have not been widely adopted. 

Surgical scoring systems may also be useful in characterizing extent of disease burden within each stage [101,198,199,200]. As previously discussed, initial disease burden has an impact on overall survival despite optimal cytoreduction, but this has only been compared between different stages. These surgical scoring systems could better characterize location as well as tumor volume within each stage and serve as a prognostic predictor. 

In November 2021, the Food and Drug Administration approved pafolacianine, which is a drug designed to aid surgeons in identifying ovarian cancer lesions intra-operatively as an adjunct to visual inspection and palpation at time of debulking surgery. This followed a phase 3 trial from 11 sites in the US and the Netherlands that showed pafolacianine identified additional lesions for resection in 33% of patients undergoing PDS and in 39% undergoing IDS [203]. This drug may offer an increase in those patients who are optimally resected. However, no survival data are available at this time.

Imaging modalities including MRI or PET /CT have been investigated as potential predictors of complete cytoreduction at time of PDS with inconsistent findings. Multiple studies have cited findings on CT associated with a decreased likelihood of complete cytoreduction including diffuse peritoneal thickening, large volume ascites, and hypermetabolic regions in the central, right upper, and left upper regions [204,205,206]. However, these findings have not been evaluated in large studies and some studies have even shown that CT is not accurate in predicting complete cytoreduction [207]. CT is currently the standard of care for pre-operative evaluation, but MRI may be useful in detecting small peritoneal deposits missed on CT [208]. MRI and CT have been compared in small studies with some showing equal accuracy in predicting optimal cytoreduction and others showing MRI may be superior, but further studies are needed to assess if MRI is predictive of optimal cytoreduction [209,210,211]. 

CA-125 has also been investigated as a predictive biomarker for optimal cytoreduction with inconsistent findings. In studies that have found CA-125 to be predictive of optimal cytoreduction, cutoffs vary, but are typically <500 with higher indicating need for consideration of NACT [212,213,214,215,216]. Other studies have found that CA-125 is not accurate in predicting optimal debulking [217,218,219]. At this point, there is insufficient evidence to rely on CA-125 as a predictive marker for complete cytoreduction. 

Another characteristic that can be used to stratify patients into treatment modalities is age. It has been shown that elderly patients over age 70 that underwent NACT followed by IDS have reduced perioperative morbidity including less blood loss at time of surgery, fewer bowel resections, shorter ICU stays, and shorter hospital stays [220]. Prior to the publication of randomized trials, there was bias when selecting which patients are to undergo PDS or NACT followed by IDS where elderly patients and those with medical comorbidities were selected for NACT, but survival has not been shown to be vastly different between groups leading to the conclusion that NACT is a reasonable alternative in those patients at high risk for surgical complications [165,189,221].

Additional studies assessing age have been performed with conflicting age cutoffs for offering NACT as a reasonable alternative including age >75 or >80 years as the lower limit of age to offer NACT over PDS [221,222]. In some studies, even some patients >80 years old do not experience a difference in complications with PDS compared to younger patients [223]. However, other studies have shown an additive effect of age and the number of radical procedures performed on complication rates [224].

Of note, in one retrospective study older patients who received NACT were less likely to undergo IDS than younger patients receiving NACT with no difference in PFS or OS. However, in patients over 70 years of age, those who underwent IDS had increased PFS [225]. At the time of consideration of IDS, candidates must not have progression of disease, extra-abdominal disease that has had a complete response or is now resectable, and performance status allows for maximal effort surgery [226].

Another patient factor influencing the decision to undergo NACT that has been investigated is obesity. However, obesity alone is not enough to influence the decision to undergo NACT over PDS [227]. 

At this time, there are not enough data for a fully algorithmic approach to selecting candidates for NACT and the decision must be considered on an individual patient level [148,228,229,230]. A combination of imaging, CA-125, laparoscopy, and patient factors can be useful to stratify patients into PDS or NACT. Those patients with suspected stage IIIC or IV disease should be evaluated by a gynecologic oncologist prior to initiation of therapy and should have a CT of the chest, abdomen, and pelvis. If the gynecologic oncologist prognosticates a low likelihood of optimal reduction to residual disease <1 cm, NACT should be initiated, but if there is a high likelihood of achieving optimal resection, PDS continues to be the recommended therapy [202].

#### 3.2.8. Trends over Time toward NACT for Advanced Disease

As data continue to emerge suggesting NACT followed by IDS may be an acceptable alternative to PDS, the implications on clinical practice must be considered. Multiple retrospective, non-randomized studies suggested that NACT with IDS may produce similar OS and PFS when compared to PDS for advanced ovarian cancer. Overall, the use of NACT has been increasing since the 2000s and OS for all patients has increased regardless of treatment modality [15].

After the presentation of the not yet published data from EORTC 55971 at the Society for Gynecologic Oncology (SGO) Annual Meetings in 2008 and 2009, Dewdney et al. performed a survey analysis regarding opinions and practice of NACT for advanced ovarian cancer in order to assess clinical practices among the members of SGO [231]. Significant findings of this study include 82% of the 339 responding members did not feel there was sufficient evidence to justify the use of NACT with IDS over PDS. Of the respondents, 60% of practitioners reported using NACT in <10% of advanced stage ovarian cancer cases. In short, despite the data published at this point in time, the majority of practitioners in the United States did not change their clinical opinions or practices in favor of NACT with IDS [231]. 

Two years later following publication of EORTC 55971, Cornelis et al. performed a similar study with the European Society of Gynecological Oncology (ESGO) assessing implementation of NACT with IDS in treatment of advanced stage IIIC and IV ovarian cancer [232]. In contrast to the mostly US respondents from 2 years prior, a majority (70.2%) of respondents felt there was sufficient evidence for the use of NACT with IDS for late-stage ovarian cancer treatment. Additionally, only 30% of respondents stated that they use NACT in <10% of their patients [232]. When respondents were asked which patients would most benefit from NACT, more than half of respondents suggested women with bulky disease in the upper abdomen, stage IV disease, medically unfit patients, or metastasis at the porta hepatis would benefit the most [232].

Of note, the self-reported rates of optimal cytoreduction with PDS differed between these two studies. On the SGO survey, 39% of respondents reported >80% success rate in optimal cytoreduction with PDS compared to only 5.5–13.8% (depending on definition of optimal debulking) on the ESGO survey reporting this success rate [231,232]. This difference in perspective on the likelihood of achieving optimal debulking with PDS may contribute to the acceptance and adoption of NACT as primary treatment. 

In 2017, following the publication of CHORUS in 2015, Huelsmann et al. performed a 5-year follow up of the survey of SGO members performed by Dewdney et al. in order to assess if opinions had changed over time with regard to the use of NACT with IDS as primary treatment in advanced ovarian cancer [231,233]. At this time, 68% of respondents (compared to 82% in 2010) still did not consider available evidence sufficient to justify use of NACT with IDS and 79% felt it should not be the preferred treatment. However, while in 2010 60% of respondents used NACT <10% of the time, in 2017, 25% of respondents used NACT < 10% of the time. While PDS continued to remain the preferred treatment at this time, a gradual acceptance of NACT with IDS as an alternative to PDS was occurring [233].

This trend toward acceptance of NACT has also been shown in other non-randomized studies. In 2016, Meyer et al. published a multicenter observational study of 1538 women with stage IIC and IV EOC comparing NACT to PDS. The authors found that NACT use increased from 16% in the 2003 to 2010 timeframe to 34% during the 2011 to 2012 time frame in stage IIIC disease and from 41% to 62% in stage IV disease [234]. Melamed et al. found similar results in a retrospective review of the NCCN database published in 2016 looking at rates of treatment modality in 40,694 women with stage IIIC and IV EOC between 2004 and 2013. They found that the proportion of women treated with NACT and IDS increased from 8.6% in 2004 to 22.6% in 2013 with the largest increase in rates of this treatment modality occurring after 2007 [12].

Interestingly, following publication of EORTC 55971, there was a transient decrease in extended cytoreductive procedures including ileostomy or colostomy formation and resection of colon, small intestine, liver, diaphragm, spleen, and stomach from 2010 to 2011. However, the rate of these procedures rose again from 2012 to 2013 [235].

In 2021, Melamed et al. went on to publish another large retrospective NCDB database study analyzing trends in OS as well as early mortality rates over time between programs that had a low use of NACT and programs that had a high use of NACT in treatment of stage IIIC and IV ovarian cancer [236]. This study analyzed use of NACT before the publication of EORTC 55971 in 2010 and after 2010. Low-use programs used NACT for advanced ovarian cancer treatment in 20.1% of cases prior to 2010 and 22.5% of the time post 2010. In contrast, high-use programs increased the percentage of patients treated with NACT from 21.7% prior to 2010 to 42.2% post 2010 [236]. Median OS improved across the entire cohort after 2010 where OS in the low-use group improved from 31.4 months from 2004–2009 to 36.8 months from 2010 to 2015 and 31.6 months to 37.9 months in the high-use group [236]. Additionally, patients treated in high-use programs had greater reductions in early, 30-day, and 90-day postoperative mortality over this time period suggesting increased use of NACT led to decreases in postoperative complications and deaths without a negative effect on OS [236]. This reduction in 30 day and 90 day postoperative mortality was also seen in another large NCDB database study performed in 2019, which also showed an increase in overall complexity of surgery regardless of initial treatment modality [237].

## 4. Conclusions

Ovarian cancer is a deadly disease accounting for a disproportionately high number of cancer-related deaths in women compared to its incidence, and optimal initial treatment of advanced ovarian cancer has remained a subject of debate. While PDS has remained the mainstay of treatment since the 1970s, the survival benefit has only been seen with optimal resection which can be challenging in cases with high disease burden and in high-risk surgical candidates with multiple comorbidities. Even still, survival following PDS in those optimally debulked remains less than ideal. 

NACT followed by IDS has emerged as an alternative treatment modality with the goal of improving survival past what PDS has offered patients in the past in terms of survival. However, to date, there have been four randomized controlled trials comparing NACT to PDS with inconclusive results where two demonstrated non-inferiority, one failed to demonstrate non-inferiority, and one failed to demonstrate superiority. While non-inferiority or superiority have not been confirmed, NACT followed by IDS has been shown to have fewer postoperative adverse events and is a reasonable alternative in select patient populations including the elderly, those with multiple comorbidities, poor surgical candidates, and those with extensive metastases resulting in decreased morbidity without a negative impact on survival. 

Through these studies assessing NACT, optimal debulking has continued to remain a significant factor in survival whether achieved with PDS or NACT followed by IDS. In some studies, NACT followed by IDS has resulted in higher rates of optimal debulking and therefore increased survival. While there are tools to gauge the likelihood of optimal debulking such as the Fagotti score and Sugarbaker’s PCI, no tool exists that takes the entire clinical picture into account to aid the clinician in selecting appropriate candidates for PDS versus NACT. In the future, we should strive for a strategy such as a risk calculator or nomogram that would provide standardized guidance to consider in this complex decision making. 

While trends have shown NACT is becoming more widely accepted amongst practicing gynecologic oncologists as an alternative to PDS, further randomized controlled trials (preferably all of one trial design, i.e., all inferiority or all superiority) are needed to assess the efficacy as well as long-term impact of NACT as primary treatment. Efforts should be undertaken to exclude low-grade serous carcinomas given different biology, create uniform and acceptable strategies to identify correct candidates for NACT including use of laparoscopy for pre-op planning and adequate tissue acquisition, and exclude those who have histology proven endometrial cancer. At this time, the current data in the literature support the use of NACT as an alternative to PDS in a select patient population and should be considered for any patient, especially those with extensive disease unlikely to be optimally resected or those of poor surgical candidacy.

## Figures and Tables

**Figure 1 diagnostics-12-00988-f001:**
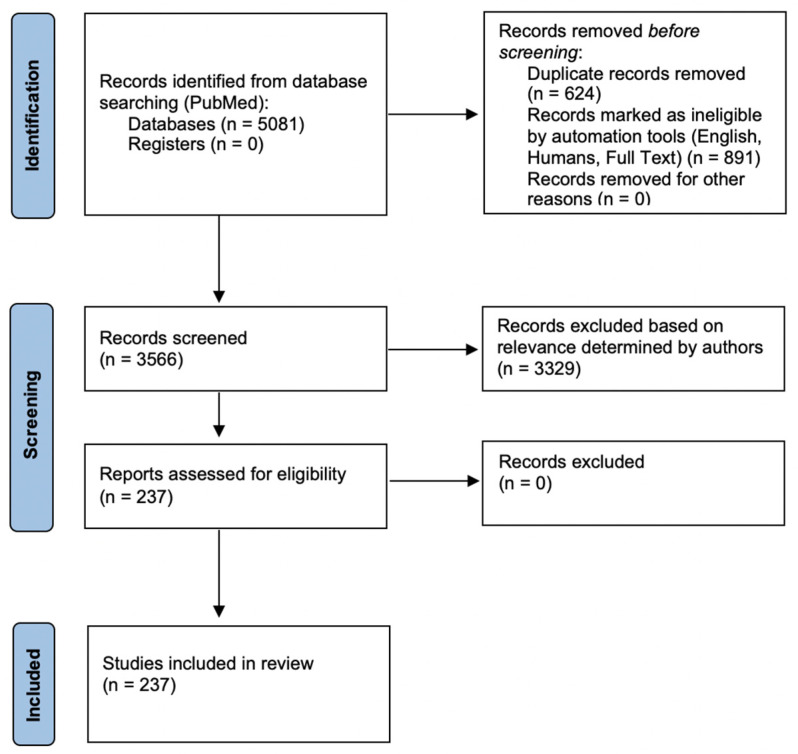
PRISMA flow diagram depicting methodology.

**Table 1 diagnostics-12-00988-t001:** FIGO staging of ovarian cancer, 2014 [5].

**Stage I** Tumor confined to ovaries	IA		Tumor involving 1 ovary Capsule intact No tumor present on external surface No malignant cells in ascites or peritoneal washings
	IB		Tumor involving both ovaries Capsule intact No tumor present on external surface No malignant cells in ascites or peritoneal washings
	IC		Tumor limited to 1 or both ovaries
		IC1	Surgical spill
		IC2	Capsule rupture before surgery or tumor on ovarian surface
		IC3	Malignant cells in ascites or peritoneal washings
**Stage II** Tumor involves 1 or both ovaries with pelvic extension (below the pelvic brim) or primary peritoneal cancer	IIA		Extension and/or implant on uterus and/or fallopian tubes
	IIB		Extension to other pelvic intraperitoneal tissues
**Stage III** Tumor involves 1 or both ovaries with peritoneal metastases outside the pelvis or retroperitoneal lymphadenopathy	IIIA		Positive retroperitoneal lymph nodes and/or microscopic metastasis beyond the pelvic brim
		IIIA1	Positive retroperitoneal lymph nodes only
			IIIA1(i) Metastasis ≤ 10 mm
			IIIA1(ii) Metastasis > 10 mm
		IIIA2	Microscopic, extra-pelvic (above the pelvic brim) peritoneal involvement ± positive retroperitoneal lymph nodes
	IIIB		Macroscopic, extra-pelvic peritoneal metastasis ≤ 2 cm ± positive retroperitoneal lymph nodes Extension to capsule of liver or spleen
	IIIC		Macroscopic, extra-pelvic peritoneal metastasis ≥ 2 cm ± positive retroperitoneal lymph nodes Extension to capsule of liver or spleen
**Stage IV** Distant metastasis excluding peritoneal metastasis	IVA		Pleural effusion with positive cytology
	IVB		Hepatic and/or splenic parenchymal metastasis, metastasis, metastasis to extra-abdominal organs (including inguinal lymph nodes and extra-abdominal lymph nodes)

**Table 2 diagnostics-12-00988-t002:** Summary of baseline characteristics of patients included in randomized trials comparing NACT to PDS. [13,14,90,91].

Characteristic	EORTC 55971		CHORUS		JCOG0602		SCORPION	
	PDS (n= 336)	NACT (n = 334)	PDS (n= 276)	NACT (n = 274)	PDS (n = 149)	NACT (n = 152)	PDS (n = 84)	NACT (n = 87)
Median Age (years)	62	63	66	65	59	60.5	54.8	56.2
FIGO Stage								
III			206	412	100	105		
IIIC	257	253					71	79
IV	77	81					13	8
Grade ^1^								
G1	14	10	13	12			1	1
G2	57	41	43	27			2	2
G3	145	130	165	149			80	84
Unknown	120	153	34	31			1	0
Histology								
Serous	220	194	25	26	115	102		
High Grade Serous			184	150			81	86
Low Grade Serous			10	9			1	1
Clear Cell	6	4	4	13			1	0
Mucinous	8	11	2	4	2	2		
Endometrioid	11	5	11	5	6	4		
Undifferentiated	69	90						
Performance Status								
WHO ^2^ 0	153	147	83	88				
WHO 1	141	143	138	133				
WHO 2	40	44	53	49				
WHO 3			1	4				
ECOG ^3^ 0–1					130	131	75	80
ECOG ≥2					19	21	9	7

^1^ Grade not reported in JCOG0602, ^2^ World Health Organization performance status, ^3^ Eastern Cooperative Oncology Group Performance Status.

**Table 3 diagnostics-12-00988-t003:** Comparison of randomized trials comparing NACT to PDS.

Study	Inclusion Criteria	Arms	Endpoints	Design	Patients	Results	Conclusion
EORTC 55971 [13]	Stage IIIC or IV WHO 0–2 ^1^	PDS: surgery → 6 cycles chemo NACT: 3 cycles chemo → surgery → 3 cycles chemo	Primary: OS Secondary: AE ^2^, QOL ^3^, PFS	Non-inferiority	670 patients 59 institutions PDS: 336 NACT: 334	PDS OS: 29 mos. NACT OS: 30 mos.	NACT is non-inferior to PDS
CHORUS [14]	Stage III or IV	PDS: surgery → 6 cycles chemo NACT: 3 cycles chemo → surgery → 3 cycles chemo	Primary: OS Secondary: PFS, QOL ^3^	Non-inferiority	550 patients 87 institutions PDS: 276 NACT: 274	PDS OS: 22.6 mos. NACT OS: 24.1 mos.	NACT is non-inferior to PDS
JCOG0602 [90]	Stage III or IV ECOG 0–3 ^4^	PDS: surgery → 8 cycles chemo NACT: 4 cycles chemo → surgery → 4 cycles chemo	Primary: OS Secondary: PFS	Non-inferiority	301 patients 34 institutions PDS: 149 NACT: 274	PDS OS: 49 mos. NACT OS: 44.3 mos.	Non-inferiority of NACT to PDS not confirmed
SCORPION [91]	Stage IIIC or IV ECOG 0–2 ^4^	PDS: surgery → 6 cycles chemo NACT: 3–4 cycles chemo → surgery → 2–3 cycles chemo for 6 total	Primary: perioperative morbidity, PFS Secondary: OS, QOL ^3^	Superiority	171 patients 1 institution PDS: 84 NACT: 87	PDS PFS: 15 mos. NACT PFS: 14 mos. PDS OS: 41 mos. NACT OS: 43 mos.	NACT is not superior to PDS Lower postop complications with NACT

^1^ World Health Organization performance status, ^2^ Adverse effects, ^3^ Quality of life, ^4^ Eastern Cooperative Oncology Group Performance Status.

## Data Availability

No new data were created or analyzed in this study. Data sharing is not applicable to this article.

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
