# Peer review of "Surgery in Advanced Ovary Cancer: Primary versus Interval Cytoreduction"

_diagnostics, 2022, doi:10.3390/diagnostics12040988_

Round 1

Reviewer 1 Report

The article is a review of a burning topic, such as what initial treatment is optimal in patients with advanced EOC. Provides a historical perspective on the evolution of PDS and NACT followed by IDS. In the conclusion, it is indicated that the most important prognostic factor for survival is optimal debulking, whether it is achieved by PDS or NACT + IDS, and that the problem lies in the fact that there is no tool that assesses the entire clinical case and helps the clinician to select candidates for PDS or NACT + IDS in order to achieve optimal debulking.

-The absence of methodology used in the review is striking. I think it would be more appropriate to take the PRISMA systematic review guidelines (Moher D, Liberati A, Tetzlaff J, Altman DG; PRISMA Group.
Preferred reporting items for systematic reviews and meta-analyses:
the PRISMA statement. PLoS Med 2009;6:e1000097) and determine how each article was selected
-Although in historical perspective the term PDS is adequate, at the end of this part it should be changed to a more modern term such as optimal cytoreduction. The term debulking seems obsolete and is associated with macroscopic residue in the surgical field.
-The term cytoreduction is associated with microscopic residue in the surgical field, so it does not seem appropriate to say R0 resections (no microscopic residue), but R1 (microscopic residue) or CC0 following the Sugarbaker CC Score (Sugarbaker PH. Technical handbook for the integration of cytoreductive surgery and perioperative intraperitoneal chemotherapy into the surgical management of gastrointestinal and gynecologic malignancy. 4th ed. Grand Rapids, Michigan: The Ludann Company; 2005, pp. 52–6.)
- Description of changes in the FIGO 2014 classification with respect to the previous one seems out of the focus of the article. It could be limited to stage III and IV. Stage IIIA2 is missing in the FIDO classification table
-line 30: ... 21,410 women will be diagnosed; I should say… 21,410 women have been diagnosed
-line 223 discusses results of "secondary debulking surgery" after non-optimal PDS; I think the more appropriate term would be "interval surgery"
-Tumor biology: the importance of the stage of the disease in the patient's prognosis is considered. The analysis of the optimal cytoreduction results must be performed within each stage and not comparing stage IIIb with stage IIIc; However, within each stage, prognostic differences based on tumor volume arise, for which more sensitive intraoperative staging methods should be used, such as the Sugarbaker PCI (Sugarbaker PH. Technical handbook for the integration of cytoreductive surgery and perioperative intraperitoneal chemotherapy into the surgical management of gastrointestinal and gynecologic malignancy. 4th ed. Grand Rapids, Michigan: The Ludann Company; 2005, pp. 52–6.) or the Fagoti laparoscopic score
-line 328: Following GOG 172, multiple studies were done to investigate IP chemotherapy regimens with decreased toxicity. GOG 9916 and 9917 substituted IP carboplatin for cisplatin ...; GOG 172 carries PI cisplatin
-Table 2.: mismatches in undifferentiated histology
-Line 477: GOG-152 in 2004 ...; This article avoids the term interval debulking surgery because it includes patients treated with maximal debulking surgery but who did not achieve optimal CR; secondary CR term is preferred here
-Line 484: it should be noted that the discussion of the article Vergote et al. published EORTC 55971
-Line 498: The patients in this study had extensive stage IIIC or IV disease with 61.6% of patients having >10 cm of metastatic disease. Again, there is a reference to a staging deficit of peritoneal disease

Reviewer 2 Report

  1. The concept of each small topic should be made clearer.
  2. The conclusion part should emphasis all the concept again.
  3. In order to minimize post-operative residual tumor volume, is there any benefit if adding bevacizumab in neoadjuvant chemotherapy regimen?
  4. About post-operative complications, what’s the difference in the four RCTs comparing NACT vs PDS in your article?
  5. After reviewing these data, in your opinion, what’s the ideal timing for NACT?
  6. A meta-analysis may be considered for all these 4 randomized trials demonstrating overall survival of PDS 681 compared to NACT in (A) EORTC 55971 [13] (B) CHORUS [14], (C) JCOG0602 [59], (D) SCOR-682 PION [60].
  7.  

Round 2

Reviewer 1 Report

It is a nice article, but I have to insist in methodology. Readers have to know criteria for selecting the references, in order to avoid bias.

The systematic review is a good option and I don´t think the article is far from it. It will give the article a more scientific and less narrative content

Round 3

Reviewer 1 Report

The article has been improved considerably

I miss the PRISMA flow chart for illustrating the followed methodology. I suggest to included it.

Author Response

This manuscript is a resubmission of an earlier submission. The following is a list of the peer review reports and author responses from that submission.